# Check the Need–Prevalence and Outcome after Transvenous Cardiac Implantable Electric Device Extraction without Reimplantation

**DOI:** 10.3390/jcm10184043

**Published:** 2021-09-07

**Authors:** Giuseppe D’Angelo, David Zweiker, Nicolai Fierro, Alessandra Marzi, Gabriele Paglino, Simone Gulletta, Mario Matta, Francesco Melillo, Caterina Bisceglia, Luca Rosario Limite, Manuela Cireddu, Pasquale Vergara, Francesco Bosica, Giulio Falasconi, Luigi Pannone, Luigia Brugliera, Teresa Oloriz, Simone Sala, Andrea Radinovic, Francesca Baratto, Lorenzo Malatino, Giovanni Peretto, Kenzaburo Nakajima, Michael D. Spartalis, Antonio Frontera, Paolo Della Bella, Patrizio Mazzone

**Affiliations:** 1Department of Cardiac Electrophysiology and Arrhythmology, IRCCS San Raffaele Scientific Institute, San Raffaele Hospital, Vita-Salute University, 20132 Milan, Italyfierro.nicolai@hsr.it (N.F.); marzi.alessandra@hsr.it (A.M.); paglino.gabriele@hsr.it (G.P.); gulletta.simone@hsr.it (S.G.); bisceglia.caterina@hsr.it (C.B.); lucalimite@gmail.com (L.R.L.); cireddu.manuela@hsr.it (M.C.); vergara.pasquale@hsr.it (P.V.); bosica.francesco@hsr.it (F.B.); giuliofalasconi@gmail.com (G.F.); pannone.luigi@hsr.it (L.P.); sala.simone@hsr.it (S.S.); radinovic.andrea@hsr.it (A.R.); baratto.francesca@hsr.it (F.B.); peretto.giovanni@hsr.it (G.P.); kenzabunakajima@gmail.com (K.N.); msparta@med.uoa.gr (M.D.S.); frontera.antonio@hsr.it (A.F.); dellabella.paolo@hsr.it (P.D.B.); mazzone.patrizio@hsr.it (P.M.); 2Third Clinical Department for Cardiology and Intensive Care, Klinik Ottakring, 1160 Vienna, Austria; 3Division of Cardiology, Medical University of Graz, 8036 Graz, Austria; 4Division of Cardiology, Sant’Andrea Hospital, 13100 Vercelli, Italy; m.matta26@gmail.com; 5Department of Cardiovascular Imaging Unit, IRCCS San Raffaele Scientific Institute, San Raffaele Hospital, Vita-Salute University, 20132 Milan, Italy; francescomelillo1989@gmail.com; 6Department of Rehabilitation and Functional Recovery, IRCCS San Raffaele Scientific Institute, Vita-Salute University, 20132 Milan, Italy; brugliera.luigia@hsr.it; 7Department of Cardiology, Hospital Universitario Clínico de Zaragoza, 50009 Zaragoza, Spain; toloriz@hotmail.com; 8Department of Clinical and Experimental Medicine, University of Catania, 95131 Catania, Italy; malatino@unict.it

**Keywords:** extraction, reimplantation, pacing, ICD, CRT

## Abstract

Background: after transvenous lead extraction (TLE) of cardiac implantable electric devices (CIEDs), some patients may not benefit from device reimplantation. This study sought to analyse predictors and long-term outcome of patients after TLE with vs. without reimplantation in a high-volume centre. Methods: all patients undergoing TLE at our centre between January 2010 and November 2015 were included into this analysis. Results: a total of 223 patients (median age 70 years, 22.0% female) were included into the study. Cardiac resynchronization therapy-defibrillator (CRT-D) was the most common device (40.4%) followed by pacemaker (PM) (31.4%), implantable cardioverter-defibrillator (ICD) (26.9%), and cardiac resynchronization therapy-PM (CRT-P) (1.4%). TLE was performed due to infection (55.6%), malfunction (35.9%), system upgrade (6.7%) or other causes (1.8%). In 14.8%, no reimplantation was performed after TLE. At a median follow-up of 41 months, no preventable arrhythmia-related events were documented in the no-reimplantation group, but 11.8% received a new CIED after 17–84 months. While there was no difference in short-term survival, five-year survival was significantly lower in the no-reimplantation group (78.3% vs. 94.7%, *p* = 0.014). Conclusions: in patients undergoing TLE, a re-evaluation of the indication for reimplantation is safe and effective. Reimplantation was not related to preventable arrhythmia events, but all-cause survival was lower.

## 1. Introduction

Cardiovascular implantable electronic devices (CIED) are increasingly used for the treatment of brady- and tachy-arrhythmic cardiomyopathies, leading to rising numbers of patients with CIEDs [1]. However, the incidence of CIED-related complications is not negligible and in some situations transvenous lead extraction (TLE) is indicated. Infection is the most feared complication, with an incidence of 1.9 per 1000 device-years [2], being responsible for relevant morbidity and potentially life-threatening complications [2,3]. Other indications for TLE include lead failure associated with adverse arrhythmic effects, vein stenosis/occlusion, presence of recalled leads, or facilitation of MRI conditionality. Furthermore, lead extraction may be considered after shared-decision making with the patient, for example during device upgrade [2].

TLE carries a non-negligible risk of procedure-related complications, such as cardiac tamponade, tricuspid valve regurgitation, embolization, vascular complications, and death [4]. Moreover, reimplantation of CIEDs after extraction puts the patient at risk of repeat infection or complications. For this reason, current guidelines recommend patients’ re-evaluation after explant, aiming to identify patients strictly requiring device reimplantation and those who can benefit from a conservative management [2,3].

The aim of this study was to identify patients without reimplantation, to assess their long-term outcome compared to remaining patients and to document the risk of further device-related complications in reimplantation patients in a high-volume tertiary centre.

## 2. Materials and Methods

This study is a retrospective analysis of all patients undergoing TLE at the Department of Cardiac Electrophysiology and Arrhythmology, San Raffaele Hospital, Milan, Italy, between January 2010 and November 2015. The institutional ethics committee approved the analysis.

### 2.1. TLE and Post-Procedural Management

The indication of TLE was set according to current guidelines [2,3] after detailed discussion with the referring physician and the patient. All TLE procedures were performed in the electrophysiology laboratory under conscious sedation or general anaesthesia using a stepwise approach as described elsewhere [4]. All lead extractions were performed with standby cardiac surgery on-site. In case of pacemaker dependency, a standard active-fixation lead was placed in the right ventricle and connected to a temporary pacemaker by the end of the procedure.

After the procedure, patients were treated in our arrhythmia unit with continuous ECG monitoring and transthoracic echocardiography was performed.

### 2.2. Decision to Reimplant

The decision to reimplant the CIED was based on the individual indication for new CIED implantation at time of TLE according to current international guidelines [5,6], taking into account the patient’s history, clinical evaluation, frailty, Holter ECG, and echocardiogram. Second level exams, such as invasive electrophysiological study or cardiac magnetic resonance, were performed in selected patients, according to the clinical presentation. The patient’s preference was especially taken into account if the indication for CIED implantation was unclear (e.g., IIb indication for implantation) or the patient strongly denied or favoured reimplantation. Pacing-dependent patients were always implanted a new CIED, but the type of device was also reassessed before reimplantation. The main indications for reimplant in pacemaker patients were intermittent or chronic high-grade AV block or symptomatic sick sinus syndrome. In patients with previous bradycardia-tachycardia syndrome, the cardiac rhythm in the year before TLE was assessed from the CIED storage and a reimplant was omitted if the patients had been in stable atrial fibrillation without episodes of bradycardia. In previous ICD patients, reimplantation was offered in patients with a history of sustained ventricular arrhythmia and a left ventricular ejection fraction below 35%. In patients with CRT, reimplant was recommended in patients with good response to CRT therapy.

The reimplantation was performed at the ipsilateral side, directly after the lead extraction, or after at least 7 days of antimicrobial therapy and negative blood cultures, for at least 72 h at the contralateral side in patients with device infection. In selected cases without the dependency of pacing, the reimplantation was performed during a second stay in hospital a few weeks after the index procedure.

### 2.3. Follow-Up

Following reimplantation or decision to discontinue device therapy, patients were discharged and followed thereafter in our clinic after one month and at a 6–12-month interval afterwards.

### 2.4. Data Collection

All patients receiving TLE at our centre were identified using the department’s prospective TLE registry. Patients were excluded if the decision to reimplant was left to the referring centre and in case of in-hospital death before the decision was made. In case of multiple extractions, the first procedure was included as index procedure. Baseline, procedural and follow-up data, as well as complications, were retrieved from the hospital’s information system. In case of missing follow-up, patients without reimplantation were additionally contacted via telephone.

### 2.5. Endpoints

Complete procedural success was defined as the removal of all targeted leads and all lead material from the vascular space without the occurrence of any permanently disabling complication or procedure-related death. Clinical success was defined as the removal of all targeted leads and lead material from the vascular space that could oppose a risk of perforation, embolic events, or perpetuation of infection, in the absence of complications. Failure of the procedure was defined as the inability to achieve either complete procedural or clinical success, or the occurrence of any permanently disabling complication, or procedure-related death. Major complications were defined as outcomes that were life-threatening, resulting in significant or permanent disability or death, or required surgical intervention. Minor complications were defined as events related to the procedure that required medical intervention or minor surgery. Device-related complication at follow-up was defined any complication that was exclusively caused by the implanted device and required invasive interventions as result; pocket changes due to battery depletion were excluded.

### 2.6. Statistics

Patients were stratified into two groups based on reimplantation after TLE: All patients that received reimplantation during the index stay or were scheduled a reimplantation procedure at the time of discharge were summarised into the “reimplantation” group, whereas remaining patients formed the “no reimplantation” group. Continuous variables are reported as mean (standard deviation, SD) or median (interquartile range, IQR), and categorical variables as percentage (absolute number). Continuous data were compared by student’s T test or Mann-Whitney U-test as appropriate; categorical variables were compared with Fisher’s test. Multivariable analysis using logistic regression analysis was performed to assess the role of predictors of the absence of need for device reimplantation. Therefore, all baseline characteristics, as shown in Table 1 with a univariable *p* value < 0.1, were included. A two-tailed *p* value < 0.05 was considered statistically significant. All analyses were performed using R 4.0.5 (The R Project, Vienna, Austria).

## 3. Results

Out of 242 patients undergoing 246 TLE procedures during the observation period, 223 patients were included into the analysis. Remaining patients either died in hospital (*n* = 2) or the decision to reimplant was not documented or left to the referring hospital (*n* = 17).

### 3.1. Baseline Characteristics and TLE Procedure

Median age was 70 (IQR 58–76) years and 22.0% were female. Main comorbidities were reduced left ventricular ejection fraction (63.2%), arterial hypertension (53.4%), chronic kidney disease (37.7%), and atrial fibrillation (34.6%, Table 1).

Overall, 40.4% of patients had a CRT-D, 31.4% a single or dual chamber pacemaker, and 26.9% an ICD. Remaining patients had a CRT-P (1.4%).

Infection was the main reason of extraction (55.6%), which was present in the pocket in 41.3%, while systemic infection with active endocarditis was identified in 16.1%. Lead malfunction was the cause of extraction in 35.9% of patients, followed by device upgrade (6.7%) and other causes (1.8%, such as patient discomfort, Table 1). Out of 2.5 ± 0.9 present leads per patient, 2.2 ± 1.1 were planned to be explanted; an explant of the entire system was planned in 78.0%. Median lead age was 68 months with a total range of 0 to 327 months.

TLE was clinically successful in 99.6% of cases and removal was complete in 95.5%, utilizing advanced extraction tools in 35.9% of cases. Both major and minor complications occurred in 3.1% each. Further details can be found in Appendix A.

### 3.2. Decision Not to Reimplant

In 34 patients (14.8%), the decision not to reimplant the CIED was taken. This included 12 patients (36.4%) that previously had an ICD, 11 patients (33.3%) with CRT-D and 10 patients (30.3%) with a pacemaker. The decision was based on a negative electrophysiological study in 21.2%, restoration of LV function in 21.2%, absence of arrhythmia during continuous ECG monitoring in 18.2%, patients’ preference in 12.1% and negative MRI in 6.1%. Persistent infection was another factor that played a role in the decision in 33.3% of cases. Another reason was negative electro-anatomical mapping in the presence of ARVD (*n* = 1, 3.0%). More details about patients that did not receive a reimplantation can be found in Appendix A.

In patients with reimplantation, a device upgrade was performed in 14.2%, while the device was downgraded in 9.0%. The reimplantation was performed mostly on the contralateral side (55.8%). In one case (0.5%), the reimplantation was performed with epicardial leads with the device in the abdomen.

### 3.3. Factors Favouring No Reimplantation

In patients without reimplantation, a reduced left ventricular ejection fraction was more prevalent (Table 1). Regarding the indication for CIED implantation, there were significant differences between groups, with sick sinus syndrome and inherited cardiac disease being more common in patients without reimplant. Device infection was significantly more common in this patient group (81.8% vs. 51.1%, *p* = 0.001), especially presence of endocarditis (33.3% vs. 13.2%, *p* = 0.008). In multivariable analysis, absence of ischemic cardiomyopathy (*p* = 0.047) and absence of AV block (*p* = 0.014) were significant predictors for absence of reimplantation, as well as high left-ventricular ejection fraction (*p* = 0.024, Table 2).

### 3.4. Follow-Up

Median follow-up duration was 42 months with no differences between groups (Table 3). While cumulative one-year mortality in patients with vs. without reimplantation was similar (98.0% vs. 100.0%), five-year mortality was significantly higher in patients with reimplantation (94.7% vs. 78.3%, *p* = 0.014, Figure 1). Hospitalizations for device revision (in the reimplantation group) or late reimplantation (in the no-reimplantation group) were similar (11.1% vs. 12.1%, *p* = 0.771)

In patients with reimplantation, device-related hospitalizations (excluding pocket changes due to battery depletion) occurred in 11.1% after an interval of 27 ± 25 months after the extraction procedure (with a range of 1 to 84 months). Reasons were lead failure necessitating repositioning (7.9%, mostly due to dislocation), pocket revision in imminent decubitus (2.1%), and generator recall (1.1%). There was one case of recurrent device infection in a female CRT-D patient that initially received extraction due of a 94-month-old fractured right ventricular lead. During follow-up after 77 months, she developed pocket dehiscence that progressed to pocket infection despite two surgical pocket revisions. Finally, a second CIED extraction procedure was performed in six patients (3.2%) a median of 42 months after the procedure.

In patients without reimplantation, 15.2% received an implantable loop recorder for detection of pauses in two patients (6.1%) with previous PM and for detection of ventricular arrhythmias in three patients (9.1%) with previous ICD. In total, four patients (12.1%) had a late reimplantation of their device; two patients received an ICD for ventricular arrhythmias and two patients received a CRT-D reimplantation for progressing heart failure. No repeat hospitalizations for invasive treatment of recurrent infection were documented (Appendix A).

## 4. Discussion

This analysis of consecutive patients undergoing TLE at our centre reveals that (1) prevention of reimplantation was possible in a significant proportion of patients undergoing TLE with a low risk of arrhythmia-related events; (2) baseline comorbidities and the primary indication for CIED implantation are the main predictors for device reimplantation; and (3) long-term mortality was higher in patients without reimplantation, but mostly due to non-cardiac causes.

This study shows that following a rigorous work-up, patients that do not profit from a CIED reimplantation can be identified with a low risk of complications due to undertreatment. In this analysis, in 15.2% of cases an immediate reimplantation was prevented. Due to the heterogeneity of clinical characteristics of patients undergoing TLE, we did not identify a “one size fits all” regimen to evaluate the need for reimplantation; instead, a patient-tailored approach was necessary, including the patient’s clinical status and will, cardiac magnetic resonance imaging, electrophysiologic study, and loop recorder implantation. As a CIED implantation may has a significant impact on the patient’s daily life [7], the patient’s opinion has to be incorporated in the final decision; it played a major role in 14.8% of cases without reimplantation in this analysis. In multiple regression analysis, we identified the CIED indication and the current left-ventricular ejection fraction to be significantly correlated with the decision to reimplantation. Patients without high degree atrioventricular block were more likely to be discharged without a CIED, probably because other indications for PM implantation have a higher potential to resolve (e.g., permanent AF in patients with previous symptomatic brady-tachy-syndrome). Patients without reimplantation were less likely to suffer from ischemic cardiomyopathy and reduced left-ventricular ejection fraction, as these conditions represent a class I indication for ICD implantation according to current guidelines [5]. In the current literature, similar (14.3%) [8] or even higher rates (40.7%) [9] of patients without reimplantation after TLE can be found. Differences in reimplantation may be explained by the incorporation of patients receiving TLE for indications other than CIED infection in this analysis. Interestingly, we did not find patients that had no indication for CIED therapy at time of implantation in contrast to Döring et al., who reported a proportion of 27% in patients without reimplantation [9].

During follow-up, we fortunately did not document a signal towards events caused by missing reimplantation in the no-reimplant group and for 17 months, no reimplantation occurred. Furthermore, we did not document ongoing device complications leading to repeat interventions in this group. Considering long-term outcome, it is apparent that more than one in ten patients out of this group may still need a reimplantation, but many years after initial TLE. Therefore, medical checks at regular intervals may be good for these patients, which is indeed more difficult considering that they do not have a CIED anymore. Interestingly, hospitalizations for CIED revision in the reimplantation group and late CIED reimplantation in the no-reimplantation group were similar.

In the whole population, there was no reintervention due to recurrent CIED infection necessary; only one patient with previous lead failure developed device infection at follow-up. The low rate of CIED reinfection is consistent with previous literature [10]. However, a significant proportion of patients with device-reimplantation (7.9%) had to be hospitalised for CIED revision, mostly due to lead dislocation. Therefore, a close follow-up may be beneficial in these patients.

While long-term survival was significantly lower in the no-reimplantation group, we did not identify deaths that may have been prevented by CIEDs. The reduced mortality in the no-reimplantation group was also seen in other studies dealing with reimplantation after TLE [8,9], but this effect is explained to be caused by older age and a high rate of non-cardiac deaths in the no-reimplantation group [9].

### Limitations

While this study adds valuable evidence on the long-term outcome of patients undergoing TLE at a high-volume centre with vs. without reimplantation, it is subject to a few limitations: first, it is subject to bias (such as information bias) due to its retrospective nature. Despite rigorous investigation and telephonic contact of patients, some patients were lost to follow-up. Second, the low rate of patients may have led to underpowering of factors that explain no-reimplantation. Third, this analysis may not be extrapolated to other centres with a different volume and different TLE indications as well as procedures. Furthermore, no data about dependency on temporary pacing after TLE was available, as well as details of the primary CIED implantation (e.g., LVEF in CRT patients). Lastly, with the evolution of leadless pacing in the last years new concepts may allow the reimplantation of devices that previously was deemed too risky [11,12].

## 5. Conclusions

The prevention of reimplantation after TLE, after careful evaluation, is safe and does not lead to an increased rate of preventable arrhythmia-related events. The primary indication and current left-ventricular ejection fraction represent independent predictors for reimplantation. Patients without reimplantation experience reduced long-term survival compared to remaining patients at follow-up.

## Figures and Tables

**Figure 1 jcm-10-04043-f001:**
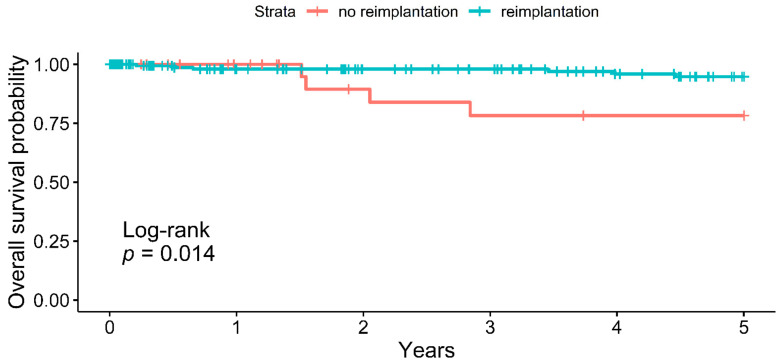
Cumulative survival of no-reimplantation vs. reimplantation groups.

**Table 1 jcm-10-04043-t001:** Baseline characteristics of the included population.

	Total Population (*n* = 223)	Reimplantation (*n* = 190)	No Reimplantation (*n* = 33)	*p*-Value
**Demographics**				
Age (years)	70 (58–76)	70 (58–76)	73 (57–78)	0.703
Female gender	22.0% (*n* = 49)	23.2% (*n* = 44)	15.2% (*n* = 5)	0.369
Comorbidities				
Hypertension	53.4% (*n* = 119)	52.6% (*n* = 100)	57.6% (*n* = 19)	0.706
Diabetes mellitus	22.0% (*n* = 49)	20.5% (*n* = 39)	30.3% (*n* = 10)	0.254
eGFR	69.7 ± 27.7	68.7 ± 27.2	75.3 ± 30.0	0.245
eGFR < 60 mL/min	37.7% (*n* = 84)	38.4% (*n* = 73)	33.3% (*n* = 11)	0.698
LVEF				0.043
35–50%	28.7% (*n* = 64)	29.0% (*n* = 55)	27.3% (*n* = 9)	
<35%	34.5% (*n* = 77)	37.4% (*n* = 71)	18.2% (*n* = 6)	
Atrial fibrillation				0.613
paroxysmal	22.9% (*n* = 51)	22.1% (*n* = 42)	27.3% (*n* = 9)	
permanent	11.7% (*n* = 26)	11.1% (*n* = 21)	1.52% (*n* = 5)	
Anticoagulation	31.4% (*n* = 70)	32.1% (*n* = 61)	27.3% (*n* = 9)	0.686
Antiplatelets	33.2% (*n* = 74)	34.7% (*n* = 66)	24.2% (*n* = 8)	0.317
**Device details**				
Device type				0.590
CRT-D	40.4% (*n* = 90)	41.6% (*n* = 79)	33.3% (*n* = 11)	
PM	31.4% (*n* = 70)	31.6% (*n* = 60)	30.3% (*n* = 10)	
ICD	26.9% (*n* = 60)	25.3% (*n* = 48)	36.4% (*n* = 12)	
CRT-P	1.4% (*n* = 3)	1.6% (*n* = 3)	0% (*n* = 0)	
Indication for implant				0.004
Non-ischemic CMP	38.1% (*n* = 85)	37.9% (*n* = 72)	39.4% (*n* = 13)	
Ischemic CMP	29.6% (*n* = 66)	32.1% (*n* = 61)	15.2% (*n* = 5)	
AV block	12.6% (*n* = 28)	14.2% (*n* = 27)	3.0% (*n* = 1)	
Sick sinus syndrome	11.2% (*n* = 25)	8.4% (*n* = 16)	27.3% (*n* = 9)	
Inherited cardiac disease	5.4% (*n* = 12)	4.7% (*n* = 9)	9.1% (*n* = 3)	
other	3.1% (*n* = 7)	2.6% (*n* = 5)	6.1% (*n* = 2)	
Implant for secondary prevention ^†^	15.5% (*n*= 16)	17.2% (*n* = 15)	6.3% (*n* = 1)	0.456
Indication for explant				0.003
infection	55.6% (*n* = 124)	51.1% (*n* = 97)	81.8% (*n* = 27)	
malfunction	35.9% (*n* = 80)	39.5% (*n* = 75)	15.2% (*n* = 5)	
system upgrade	6.7% (*n* = 15)	7.9% (*n* = 15)	0% (*n* = 0)	
other causes	1.8% (*n* = 4)	1.6% (*n* = 3)	3.0% (*n* = 1)	
Number of leads	2 (2–3)	2 (2–3)	2 (2–3)	0.943
Number of leads to be removed	2 (1–3)	2 (1–3)	2 (2–3)	0.033
Age of device (months)	54 (21–85)	50 (21–84)	68 (20–92)	0.633
Age of leads (months)	68 (31–100)	62 (30–99)	78 (45–104)	0.432

AV: atrioventricular; eGFR: estimated glomerular filtration rate calculated with CKD-EPI formula; LVEF: left ventricular ejection fraction; ICD: implantable cardioverter defibrillator; CMP: cardiomyopathy; CRT-D: cardiac resynchronization therapy-defibrillator; CRT-P: cardiac resynchronization therapy-pacemaker; PM: pacemaker; ^†^ secondary prevention as indication for primary CIED implantation in ICD and CRT-D patients (data available in 69% of cases).

**Table 2 jcm-10-04043-t002:** Univariable and multivariable analysis assessing the role of clinical parameters in predicting the absence of reimplantation.

Parameter	*p* Value (Univariable)	OR (95% CI)	*p* Value (Multivariable)
**Details regarding CIED indication**			
Absence of ischemic CMP	0.062	3.1 (1.1–10.4)	0.047 *
Absence of AV block	0.089	14.6 (2.5–281.7)	0.014 *
Sick sinus syndrome	0.004 *	1.6 (0.5–5.2)	0.425
**Clinical details**			
LVEF (per 10% increase)	0.019 *	1.5 (1.1–2.3)	0.024 *
**Details regarding CIED explant**			
Absence of lead malfunction	0.006 *	2.2 (0.3–14.4)	0.416
Number of explanted leads, per lead	0.033 *	1.3 (0.8–2.2)	0.243
CIED infection	0.001 *	2.3 (0.4–18.5)	0.373

*: *p* < 0.05; CIED: cardiac implantable electric device; CMP: cardiomyopathy; AV: atrioventricular; LVEF: left ventricular ejection fraction; OR: odds ratio.

**Table 3 jcm-10-04043-t003:** Follow-up.

	Reimplantation (*n* = 190)	No Reimplantation (*n* = 33)	*p* Value
Follow-up duration, months	44 (7–76)	23 (11–80)	0.883
1-year cumulative survival (NaR)	98.0% (131)	100.0% (23)	0.500
5-year cumulative survival (NaR)	94.7% (73)	78.3% (13)	0.014 *
late reimplantation or device revision	11.1% (*n* = 21)	12.1% (*n* = 4)	0.771
**“Reimplantation”-specific events**		N/A	N/A
any device-related hospitalisation	11.1% (*n* = 21)		
lead failure/dislocation	7.9% (*n* = 15)		
pocket revision	2.1% (*n* = 4)		
device recall	1.1% (*n* = 2)		
device infection	0.5% (*n* = 1)		
repeat extraction procedure	3.2% (*n* = 6)		
**“No reimplantation”-specific events** Reimplantation	N/A	12.1% (*n* = 4)	N/A

NaR: number at risk; *: *p* < 0.05.

## Data Availability

The data presented in this study are available on request from the corresponding author.

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
