# Peer review of "Check the Need–Prevalence and Outcome after Transvenous Cardiac Implantable Electric Device Extraction without Reimplantation"

_jcm, 2021, doi:10.3390/jcm10184043_

Round 1

Reviewer 1 Report

D'Angelo, Zweiker, and their colleagues reported that in groups with CIED implantation and without CIED reimplantation after TLE at high-volume center, the predictors of CIED reimplantation, long-term outcomes, and risk of device-related complications.

Their work is worth reporting in the sense of calling attention to uniform CIED reimplantation after TLE.

However, there are some important improvements in this study.

Major

  1. Ambiguous criteria for not reimplantation of CIED

The objective criteria for not re-implanting CIEDs should be indicated, for example, show the cut-off value of LVEF in UCG or scar volume in MRI etc.

In addition, what was emphasized and how much was emphasized among various evaluation criteria (patient's history, clinical evaluation, Holter ECG, echocardiogram and the patient's preference).

The process of making the final decision should be clarified because the decision to re-implant was biased by each operator due to a retrospective study.

  1. Mismatch of comparison cohorts

Reimplantation group included patients who were pacing-dependent and who absolutely needed CIED reimplantation. If the author wanted to compare the need for CIED reimplantation between the two groups, these patients should be excluded and the backgrounds of the no reimplantation group and reimplantation group should match as closely as possible.

  1. Event rate in reimplantation group and no reimplantation group

The incidence of device-related complications in the reimplantation group (14.8%, observation period 1-84 months) and the rate requiring reimplantation in the no-implantation group (11.8%, observation period 17-108 months) appear to be numerically similar.

Although there was a comparison between the two groups in overall survival, the analysis including device related complications and the need for reimplantation was also required.

  1. Indication at the first CIED implantation

Table 1 and Supplemental table 2 described basic heart disease or arrhythmia for the initial CIED implantation, but did not mention the specific indications.

Please report whether primary prevention or secondary prevention in the case of ICD, and the value of LVEF before implantation for CRT cases.

Minor

One S-ICD patient should be excluded from the analysis as it was a validation of TLE cases.

Author Response

D'Angelo, Zweiker, and their colleagues reported that in groups with CIED implantation and without CIED reimplantation after TLE at high-volume center, the predictors of CIED reimplantation, long-term outcomes, and risk of device-related complications.

Their work is worth reporting in the sense of calling attention to uniform CIED reimplantation after TLE.

- We thank the reviewer for the appreciation of the manuscript!

However, there are some important improvements in this study.

Major

  1. Ambiguous criteria for not reimplantation of CIED

The objective criteria for not re-implanting CIEDs should be indicated, for example, show the cut-off value of LVEF in UCG or scar volume in MRI etc.

In addition, what was emphasized and how much was emphasized among various evaluation criteria (patient's history, clinical evaluation, Holter ECG, echocardiogram and the patient's preference).

The process of making the final decision should be clarified because the decision to re-implant was biased by each operator due to a retrospective study.

- We thank the reviewer for this important comment as the decision not to reimplant the CIED after transvenous lead extraction is the most important is the cornerstone of this manuscript. In non-pacing-dependent patients, the decision was primarily based on the individual indication to perform a new implantation according to current international guidelines. In many patients we performed additional tests (such as cardiac magnetic resonance imaging, electrophysiologic study or Holter ECG). The final decision was taken incorporating the patient’s wish, especially in patients with unclear indication (e.g., IIb indications by international guidelines). We incorporated these statements into the manuscript in the Methods section.

  1. Mismatch of comparison cohorts

Reimplantation group included patients who were pacing-dependent and who absolutely needed CIED reimplantation. If the author wanted to compare the need for CIED reimplantation between the two groups, these patients should be excluded and the backgrounds of the no reimplantation group and reimplantation group should match as closely as possible.

- We want to emphasize that not only patients requiring continuous temporary pacing after TLE have the absolute necessity of CIED reimplantation. Patients with recurrent episodes of idiopathic ventricular fibrillation requiring shock every few months despite antiarrhythmic therapy and patients with intermittent high-grade AV block do also benefit clearly from device reimplantation. Furthermore, seemingly “pacing-dependent” patients may experience a recovery of intrinsic rhythm after reduction of the basal pacing rate, even if there is a history of 100% pacing during device interrogation. Thus, we believe that in many cases it cannot be clearly determined if patients needed CIED reimplantation. We believe that this study is very important to make the decision easier for many operators.

- To overcome the risk of intermittent post-procedural bradycardia after TLE, our centre was very liberal with the use of temporary pacing after TLE. Therefore, temporary pacemaker placement was very common after TLE in our cohort. We unfortunately cannot perform an additional analysis excluding pacing-dependent patients because this data is not available from our internal registry. We added a sentence describing this limitation in the “Limitations” section.

  1. Event rate in reimplantation group and no reimplantation group

The incidence of device-related complications in the reimplantation group (14.8%, observation period 1-84 months) and the rate requiring reimplantation in the no-implantation group (11.8%, observation period 17-108 months) appear to be numerically similar.

Although there was a comparison between the two groups in overall survival, the analysis including device related complications and the need for reimplantation was also required.

- We thank the reviewer for this important comment. We had not compared the proportion of patients undergoing device reimplantation with those needing device revision, but we added it now (Table 3) and commented on it in the Discussion.

  1. Indication at the first CIED implantation

Table 1 and Supplemental table 2 described basic heart disease or arrhythmia for the initial CIED implantation, but did not mention the specific indications.

Please report whether primary prevention or secondary prevention in the case of ICD, and the value of LVEF before implantation for CRT cases.

- As the reviewer correctly described, the presence of previous sustained arrhythmias plays a major role in the decision whether to reimplant a CIED in ICD patients. Due to incompleteness of source data regarding primary vs. secondary prevention in ICD patients in this registry, we did not include these data in the initial manuscript. Therefore, it was not possible to include these data into the multivariate analysis. In this revised submission, we now report data in about 70% of ICD and CRT-D patients at Table 1. Interestingly, we did not find a significant difference in secondary prevention as indication for primary CIED implantation in this analysis.

- Unfortunately, LVEF at time of CRT implantation is not available in our analysis. We added this limitation into out Limitations section.

Minor

One S-ICD patient should be excluded from the analysis as it was a validation of TLE cases.

- To include only patients undergoing TLE, we performed a complete recalculation of all analyses, excluding the S-ICD patient. We therefore performed changes in the whole manuscript (including all tables and figures).

Reviewer 2 Report

This work is from a group who provide a high volume extraction service, which shows in the numbers and reported complications.  These may be more than reported in recent cohorts but in keeping with volume and general experience.

The report highlights an important issue around the need to reassess appropriateness for on-going device support for patient's who have undergone device removal or extraction.   This aspect is often overlooked within the clinical community and the authors have described their experience.  The findings of a disparity in terms of mortality differences between those re-implanted versus not may be explained by the differences in the pre-morbid underlying indications for device support and the co-morbidities.  The role of infections and subsequent late deaths rate would be difficult to quantify easily but may have bearing on the findings.  As always consideration of frailty and the clinical gestalt of the patients need for and ability to manage with a replacement system remains problematic in terms of assessing and nuanced.  These are alluded to, as well as the efforts taken to investigate need for further device support.  The authors provide information on factors used to not re-implant - which are self explanatory.  The Supplements note the higher complication rates amongst those not re-implanted, raising the issue of whether this is a factor in the decision making that needs to be addressed.  They also provide information for the cohort about progress- which is interesting, but for a relatively short duration particularly post infection.  For those undergoing extraction and re-implantation (for infective or non-infective reasons), the authors provide evidence to support this as a successful strategy - even though the group is not stratified by presence or absence of systemic infection.

Author Response

We thank the reviewer for the positive comment on the manuscript.

The authors completely agree with the reviewer that frailty also does play a major role in the outcome of patients undergoing TLE and should therefore be incorporated in the decision whether to reimplant a CIED. In our centre, each patient is assessed individually, incorporating the patient’s specific needs. However, no specific score for frailty and mobility is documented. We adapted the manuscript, including the importance of assessing the patient’s clinical status (including his/her frailty) into the decision process whether to reimplant the CIED in the Methods and Discussion.

Round 2

Reviewer 1 Report

The authors answered my questions and comments accurately.
The manuscript has been amendment and the necessary reanalysis has been done.
There are no additional comments or suggestions regarding the resubmitted manuscripts or charts.